Methods

# RedRibbon: A new rank–rank hypergeometric overlap for gene and transcript expression signatures

Anthony Piron[1,2,3] , Florian Szymczak[1,2], Theodora Papadopoulou[1,2], Maria Inês Alvelos[1], Matthieu Defrance[2,3], Tom Lenaerts[2,3,4], Décio L Eizirik[1], Miriam Cnop[1,5]

**High-throughput omics technologies have generated a wealth of large protein, gene, and transcript datasets that have exacerbated the need for new methods to analyse and compare big datasets. Rank–rank hypergeometric overlap is an important threshold-free method to combine and visualize two ranked lists of *P*-values or fold-changes, usually from differential gene expression analyses. Here, we introduce a new rank–rank hypergeometric overlap-based method aimed at gene level and alternative splicing analyses at transcript or exon level, hitherto unreachable as transcript numbers are an order of magnitude larger than gene numbers. We tested the tool on synthetic and real datasets at gene and transcript levels to detect correlation and anticorrelation patterns and found it to be fast and accurate, even on very large datasets thanks to an evolutionary algorithm-based minimal *P*-value search. The tool comes with a ready-to-use permutation scheme allowing the computation of adjusted *P*-values at low time cost. The package compatibility mode is a drop-in replacement to previous packages. RedRibbon holds the promise to accurately extricate detailed information from large comparative analyses.**

## Introduction

During the past two decades, we have seen a democratization of high-throughput sequencing technologies. The cost of DNA sequencing went down by six orders of magnitude from 2000 to 2022 (Lewin et al, 2018). High-throughput sequencing has led to the generation of large and diverse datasets covering multiple omics, including genomes, transcriptomes, and proteomes. Alternative splicing generates massive protein diversity. Through the inclusion or exclusion of exons from pre-mRNAs, distinct mature mRNAs give birth to multiple proteins with different functions (Black, 2003). Thereby, in humans, more than 200,000 different proteins are produced from around 20,000 protein coding genes (Alvelos et al,

2018). Alternative splicing is omnipresent in eukaryotic cells, affecting 80% of protein coding genes. On average, a human gene is spliced into 4.4 transcripts. Alternative splicing is implicated in many diseases (López-Bigas et al, 2005). Its analysis is challenging as most studies and pathway databases are gene centric (Liberzon et al, 2011) and the number of transcripts or splicing events can be overwhelming. Most studies aggregate transcript expression levels at the gene level, thus losing crucial information about isoforms that may play different or even opposite roles. Collectively, the widespread use of these omics technologies has exacerbated the need for new methods to analyse and compare diverse and ever larger datasets. Multiple data aggregation initiatives collected such datasets, including, among others, Gene Expression Omnibus (Edgar et al, 2002; Barrett et al, 2013), the Genotype-Tissue Expression Project (Lonsdale et al, 2013), and the Translational Human Pancreatic Islet Genotype Tissue-Expression Resource (Alonso et al, 2021).

Rank–rank hypergeometric overlap (RRHO) has been developed to compare two lists of differentially expressed genes generated with microarray technology (Plaisier et al, 2010) and it was further improved with alternative statistics and better enrichment sets (Cahill et al, 2018). RRHO compares two labelled ranked lists of real numbers. The labels can be gene/transcript identifiers or any other unique identifier. The values can be fold changes, *P*-values, slopes or another meaningful ranked statistic. The method detects the enrichments at the extremities of the ranked lists. For example, using two lists of fold change in gene expression, it allows the construction of enriched gene sets for the four possible directions, that is, down-regulated–down-regulated, up-regulated–up-regulated, down-regulated–up-regulated, and up-regulated–down-regulated. The method proceeds by computing, for all coordinates ($i,j$) in the two compared differentially expressed gene lists, an enrichment *P*-value from the number of labels in common at the extremities up to the coordinate with the hypergeometric distribution. The coordinates with the minimal *P*-value are used to determine the most significant gene set. As a result, the RRHO method is threshold-free in the sense that the detected minimal coordinate delimits the area

[1]ULB Center for Diabetes Research, Medical Faculty, Université Libre de Bruxelles, Brussels, Belgium    [2]Interuniversity Institute of Bioinformatics in Brussels (IB2), Brussels, Belgium    [3]Machine Learning Group, Université Libre de Bruxelles, Brussels, Belgium    [4]Artificial Intelligence Lab, Vrije Universiteit Brussel, Brussels, Belgium    [5]Division of Endocrinology, Erasmus Hospital, Université Libre de Bruxelles, Brussels, Belgium

Correspondence: anthony.piron@ulb.be; miriam.cnop@ulb.be

of greatest significance without the use of any threshold. The method can thus be seen as a 2D extension of 1D Gene Set Enrichment Analysis (Mootha et al, 2003; Subramanian et al, 2005) where the gene sets are replaced by a ranked differential analysis gene list. Focused gene set testing methods, like ROAST (Wu et al, 2010), compare one differential analysis with one gene set. When applied to two differential analyses, one of the differential analyses must be treated as a gene set to focus on. This focused gene set is selected with a threshold (e.g., $P$-adjusted ≤ 0.05), and the other genes are lost to the comparison. DynaVenn is another method, highly similar to RRHO, but it cannot handle large lists (the tool is tailored for lists <1,000) nor interrogate four directions (Amand et al, 2019).

The RRHO method allowed us and others to generate meaningful comparisons of differential gene and protein expression data (Colli et al, 2020a, 2020b; Lytrivi et al, 2020; Marselli et al, 2020; Blencowe et al, 2022; Yi et al, 2022), but its application revealed shortcomings that required adaptation of the original RRHO R package (Marselli et al, 2020). Nonetheless, major shortcomings remained. First, the original R package is limited by R language real number representation (R Core Team, 2022). This representation often leads to an underflow, that is, a $P$-value that is rounded to zero below a threshold. Therefore, the zero $P$-values become indistinguishable from each other, making the detection of the minimal $P$-value impossible and lowering the accuracy of the method. Second, the execution time follows a cubic growth depending on the list length, making it unpractical for large lists. To circumvent these long run times, the original R package offers the possibility to skip some coordinates in the map, trading accuracy for performance. The recommended step size is between 100 and 500 for lists of 10,000–50,000 elements (with the number of element square root as default value) introducing a potential inaccuracy of hundreds of genes.

In recent years, methods have been developed to accurately quantify transcript expression levels, including splice variants, from RNA-sequencing (RNA-seq) reads, for example, Salmon (Patro et al, 2017), kallisto (Bray et al, 2016), and RSEM (Li & Dewey, 2011). Multiple differential analysis tools can be applied to transcript-level data (Love et al, 2018; Zhu et al, 2019). Despite the progress in transcript quantification and differential analysis methods, there are, to the best of our knowledge, no tools available to compare two distinct transcript level differential analyses without prior gene level aggregation. As the total number of transcripts quantified by RNA-seq is an order of magnitude higher than for genes, existing RRHO packages are inadequate. Plaisier et al suggested to compute corrected $P$-values by permuting samples and rerunning differential and RRHO analyses a 1,000 times (Plaisier et al, 2010). This permutation method is slow for large gene expression analyses and renders transcript expression analyses prohibitive (or inaccurate with very large step sizes).

To address the above-described unmet need for comparative analysis tools of diverse omics datasets including transcripts, we developed RedRibbon. RedRibbon is a novel hypergeometric overlap method bearing in mind performance and accuracy, introducing novel data structures and algorithms, and an all-in-one permutation method to adjust the minimal $P$-value. The improvements in performance and accuracy have been assessed using synthetic datasets

and previously reported results (Marselli et al, 2020). We applied the method to compare alternative splicing results in experimental models of diabetes, including the human EndoC-$\beta$H1 beta cell line and pancreatic islets.

# Results

## Enhanced overlap maps

We first generated synthetic dataset overlap maps to exemplify RedRibbon results and illustrate its new visual features that facilitate interpretation (Fig 1). The overlap map of two perfectly identical lists is a perfect diagonal signal from down-regulation to up-regulation (Fig 1A). The hypergeometric $P$-value gets lower as coordinates are closer to the list centres and gives the whole list as enrichment. The overlap map of one list being in perfectly reversed order of the other—that is, perfect anticorrelation of gene expression changes—follows a perfect diagonal from down–up quadrant to the up–down quadrant (Fig 1B). The $P$-values are negatively signed to distinguish them as related to anticorrelated genes. The $P$-value can be plotted with different colours in the overlap map depending on their sign (not shown here).

Two lists with four quarters of 5,000 genes going respectively and perfectly in the same direction (both down-regulated or both up-regulated in the two lists) or in opposite directions (down-regulated in the first list and up-regulated in the other, and vice-versa) result in an overlap map with perfect diagonal signals for the four quadrants (Fig 1C). The maximal log $P$-values and the permutation-adjusted $P$-values are shown for each quadrant. The horizontal and vertical dotted lines split the down-regulation and up-regulation where the log fold change is zero. In this dataset, the "zero" log fold change is at two-fifth of both lists, hence, the split point is shifted to the beginning of the lists.

For completely random synthetic data lists, the fluctuation in the map is caused by random sampling and no signal is present (Fig 1D). In this case, the overlap algorithm is unable to find any significant adjusted $P$-value and no $P$-value is shown on the map for any quadrant.

## Beyond gene-level analyses

We assessed the capacity of RedRibbon to overlap gene lists but also larger lncRNA and transcript lists, and benchmarked RedRibbon against the original R package (Fig 2). First, both packages were compared using the original method (Plaisier et al, 2010)—called in this article "grid method"—for a list of n genes with a step size of $\sqrt{n}$, aiming to assess the performance of the new data structures and intersection algorithm (see the Materials and Methods section). On an Intel Xeon Processor E5-2650 v4, RedRibbon's running time increases slowly with the list size and is below 25 s for lists of 262,144 genes, whereas the running time of the original R implementation grows steeply and is already close to 200 s for lists of 65,536 genes (Fig 2, left). The original R package is limited to gene lists as the running time is prohibitive for larger lists, whereas RedRibbon can handle lncRNA and transcripts.

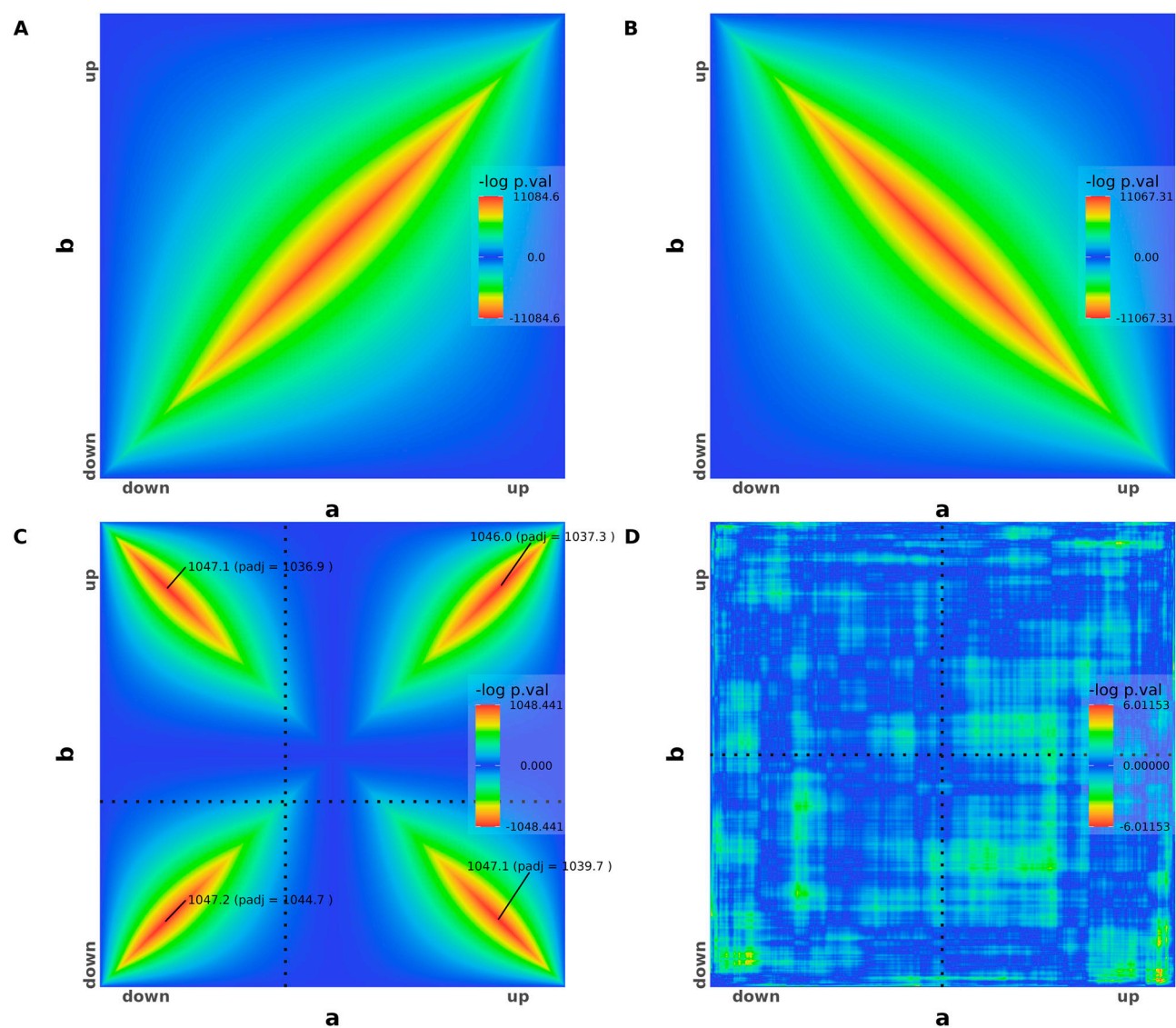

**Figure 1. Rank–rank hypergeometric overlap (RRHO) for artificial datasets.**
**(A)** RRHO map of two perfectly identical artificial gene expression lists a (on the x-axis) and b (on the y-axis). **(B)** RRHO map of two perfectly symmetrical lists with gene expression going in opposite directions. **(C)** Two lists with half of the genes going in the same direction and the other half in the opposite direction. The 96 permutation-adjusted *P*-values are reported. **(D)** RRHO map of two random gene expression lists. All adjusted *P*-values are below the significance threshold and therefore not shown (greater than 3 ≈ −*log*[0.05]).

Next, two lists of 5,000 genes and a step size of 50 were used to assess the *P*-value adjustment method, a benchmark setting used by Plaisier et al (2010). The running time they reported (8,345 s, after correction for CPU performance) is used as reference. RedRibbon outperforms this by four orders of magnitude for all tested methods: grid method, parallel execution grid, and evolutionary algorithm (Fig 2, middle). The 5,000 genes were a realistic list length in 2010, when microarray was the leading technology. Today, sequencing technologies provide lists of >50,000 elements; for these, the permutation comparison according to Plaisier et al (2010) cannot be run, the running time being prohibitive. Hence, the original permutation method is not compatible with modern sequencing datasets.

Our grid method re-implementation was then compared with the evolutionary algorithm with adjusted *P*-value computation.

For the grid method, we used a square root of the list length for the step size. Both algorithms were run in parallel mode for the adjusted *P*-value permutation computation. The grid method can handle gene and lncRNA lists (at the cost of accuracy, see the Accuracy section) and even outperforms the evolutionary algorithm for shorter lists (<20,000) but only the evolutionary algorithm can handle transcript analyses (Fig 2, right). The evolutionary algorithm is usable for the analysis of millions of elements (up to 320,000 are shown in Fig 2, right), making even exon analyses possible.

## Accuracy

True positive rate (TPR), true negative rate (TNR), and accuracy were assessed for synthetic datasets (see the Materials and Methods

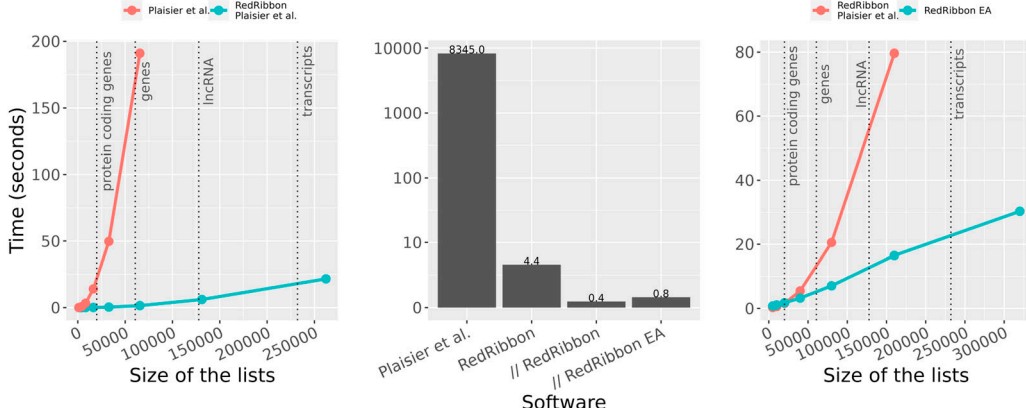

**Figure 2. RedRibbon allows performing gene, lncRNA, and transcript analyses.**
RedRibbon was compared with Plaisier et al (2010). The vertical dotted lines delimit the typical list lengths for gene, lncRNA, and transcript analyses. Left: time to compute the minimal *P*-value for lists of n elements with a step size of sqrt(n) with the Plaisier et al (2010) grid method compared with our re-implementation of this method using the same parameters. Center: comparison of our *P*-value permutation method with Plaisier et al (2010) for 5,000 elements and a step size of 50. Time according to Plaisier et al (2010) is reported and corrected for CPU performance improvement (single thread performance on https://www.cpubenchmark.net/). RedRibbon time is reported in single thread (RedRibbon), multithreads (// RedRibbon), and multithreads with the evolutionary algorithm (// EA RedRibbon). Right: time to compute the minimal *P*-value of n elements with a step size of sqrt(n) with permutation *P*-value correction. Our re-implementation of the Plaisier et al (2010) grid algorithm is compared with the new evolutionary algorithm that has no step size limitation and hence higher accuracy.

section) using RRHO (Plaisier et al, 2010), RRHO2 (Cahill et al, 2018), our reimplementation of the original RRHO method and our novel evolutionary algorithm method (Figs S1A and 3A). In our tests on synthetic datasets, RRHO and RRHO2 performed rather poorly (Fig S1A). The accuracy tests also revealed some bugs in the RRHO R package (Fig S1B) that were solved in RedRibbon (Fig S1C). The RRHO2 package only detects half of the genes (TPR = ~0.5). Although better than RHHO, the results remain rather inaccurate. RRHO2 being based on the same code as the RRHO R package, we suspect that similar bugs remained present. Hence, we focus here on our reimplementation of the grid algorithm.

Measurements showed a clear-cut advantage to the evolutionary algorithm for both TPR and accuracy (Fig 3A). In most synthetic datasets, TPR is exactly 1, meaning that all genes significantly correlated in the lists are detected, whereas for the grid method up to 11 genes in 100 are missed (TPR = ~0.89). The TNR is kept under control as for at most three genes out of 10,000 detection is missed (TNR = 0.99975) with most datasets having close to 0 misdetections. Accuracy is systematically better for the evolutionary algorithm than for the grid algorithm (Fig 3A).

We also assessed the accuracy of the grid method with increasing step size (Fig 3B). The evolutionary algorithm is step size free, and performs with near perfect TPR, TNR, and accuracy. TPR drops in the grid simulation, missing up to three genes per 100 at step size 45 (TPR = ~0.97) and 18 genes per 100 at step size 270 (TPR = ~0.83). Accuracy follows a similar pattern. The TPR and accuracy of the grid method are highly heterogenous even for step sizes very close to each other, for example, a step size of 50 has a perfect TPR, whereas a step size of 45 misses up to three genes out of 100 and similarly for step sizes of 250 and 270.

We next assessed RedRibbon on experimental datasets previously analysed by us (Marselli et al, 2020). RRHO was run between the fold changes of 16,547 genes from human islets, comparing donors with and without type 2 diabetes against islets exposed

in vitro to the saturated free fatty acid palmitate and high glucose for 48 h and subsequently allowed to recover for 4 d. The level map shows significant signals in the four quadrants with the strongest signal in the upregulated direction (Fig 3C). Comparison between the original R and RedRibbon packages shows a large intersection (Fig 3D). RedRibbon identifies 7–107 additional genes in the four quadrants of the overlap map (Fig 3D, top). The differences between the two packages result in differences in enriched pathways (Fig 3D, bottom). The extent of the differences is similar to the differences for the synthetic gene sets, suggesting the accuracy metrics are sound. The improved accuracy of RedRibbon generates new biological insight as compared with that previously reported (Marselli et al, 2020): among the 13 new pathways in the up–up overlap (Fig 3D, bottom) are regulation of transcription factor binding activity, toll-like receptor signalling, and pro-inflammatory and profibrotic mediators, pathways of importance in type 2 diabetes (Table 1) that were not detected by the original R package (Supplemental Data 1).

### Adjusted *P*-value type 1 error

To assess the soundness of the permutation method, we controlled for type 1 error against a random background composed of 1,000 random list pairs of 1,000 elements. An adjusted *P*-value below 0.05 was reported for 1.3% of the RedRibbon analyses (*P*-value = $2.5 \times 10^{-10}$ for *P*-adjusted > 0.05 null hypothesis). This below expected percentage shows that our adjustment method conservatively controls for type-1 errors.

### Alternative splicing analyses

The tool developed by Plaisier et al (2010) is not suited for splicing analyses, whereas RedRibbon has the power to analyse hundreds of thousands of transcripts (Fig 2). To validate its suitability for this type of analysis, we applied RedRibbon to previously generated

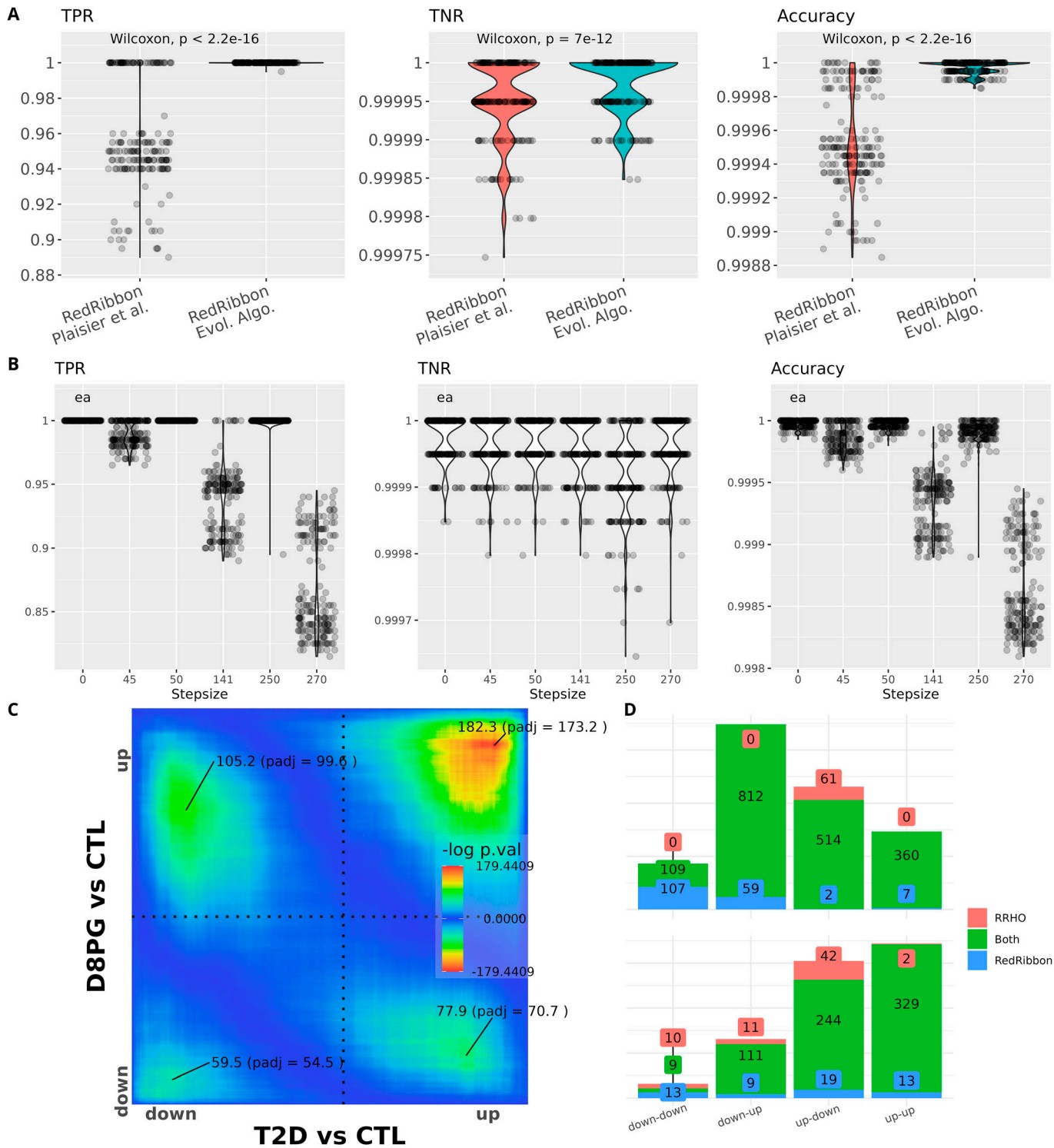

**Figure 3. RedRibbon allows performing overlap analyses with high accuracy.**
**(A)** Violin plots for 192 artificial datasets (see the synthetic gene sets and accuracy measurements section). The remaining genes are randomly ranked. True positive rate (TPR), true negative rate (TNR), and accuracy are reported. TPR and accuracy are significantly better for evolutionary algorithms than with the classic grid method. **(B)** TPR, TNR, and accuracy for the evolutionary algorithm (ea) compared with the grid method for increasing step sizes. The 141-step size is the default value for grid method. **(C)** RedRibbon hypergeometric map of 16,547 genes comparing human islets isolated from type 2 diabetic (T2D) versus nondiabetic (CTL) donors and human islets recovering from palmitate + glucose exposure in vitro (D8PG versus CTL), as in Marselli et al (2020). **(D)** Comparison of the original rank–rank hypergeometric overlap algorithm with default step size (128 genes, red) with RedRibbon (blue). Green shows elements detected with both methods. Top figure shows the overlapping gene counts, bottom figure pathway enrichment counts.

**Table 1.** 13 pathways detected by RedRibbon only in the up–up-regulation quadrant in Fig 3D.

| Pathway | P-value | Relevance to type 2 diabetes |
| --- | --- | --- |
| Regulation of protein phosphorylation | 0.012 | Batista et al (2020) |
| Regulation of osteoblast differentiation | 0.018 | |
| Regulation of DNA-binding transcription factor activity | 0.018 | Guo et al (2013); del Bosque-Plata et al (2021) |
| Regulation of receptor signaling pathway via JAK-STAT | 0.019 | Moshapa et al (2019); Wu et al (2019) |
| Wound healing | 0.021 | |
| Toll-like receptor signaling pathway | 0.022 | Sepehri et al (2016); Guo et al (2021) |
| Overview of proinflammatory and profibrotic mediators | 0.022 | Tsalamandris et al (2019) |
| Positive regulation of defense response | 0.025 | Creely et al (2007); Berbudi et al (2020) |
| Lung fibrosis | 0.028 | |
| Protein phosphorylation | 0.033 | Batista et al (2020) |
| MAPK cascade | 0.038 | Avruch (2007); He et al (2021) |
| Circulatory system development | 0.039 | |
| Regulation of cell death | 0.042 | Cnop et al (2005); Mukherjee et al (2021) |

The improved accuracy of RedRibbon detects pathways in human islets that overlap between lipoglucotoxicity and type 2 diabetes. Listed references show the importance of these pathways in type 2 diabetes.

alternative splicing data. We used our RNA-seq from human clonal EndoC-βH1 beta cells and human islets exposed to IFNα, and RNA-seq of EndoC-βH1 cells silenced for splicing factor *SRSF6* (also known as *SRp55*) (Juan-Mateu et al, 2018). *SRSF6* silencing modulates splicing of genes involved in beta cell apoptosis, JNK signalling, insulin secretion, and type 2 diabetes (Fig 4A). The exposure to IFNα induces beta cell hallmarks of type 1 diabetes, including inflammation, endoplasmic reticulum stress, and HLA class I overexpression (Marroqui et al, 2017; Coomans de Brachène et al, 2018), and is therefore disease-relevant, corresponding to early innate immune processes. IFNα also induces major alterations in splicing in beta cells (Colli et al, 2020a). Interestingly, the main *SRSF6* transcript SRSF6-201 is down-regulated by IFNα in human beta cells and islets (Supplemental Data 1). *SRSF6* may thus be responsible for some of the IFNα-induced alternative splicing. To test this hypothesis and compare the beta cell alternative splicing patterns after *SRSF6* silencing and cytokine exposure, we applied RedRibbon.

The transcript signatures of SRSF6-depleted (101,226 transcripts) versus IFNα-exposed EndoC-βH1 cells (151,157 transcripts) show substantial overlap in down- and up-regulated transcripts (Fig 4B, left panel). To perform pathway analyses, we substituted the overlapping transcript identifiers by their corresponding gene identifiers. Multiple transcripts in the same quadrant or in different quadrants may be mapped to the same gene. The loss of this multiplicity information is inherent to using gene-centric pathway databases and will be solved when transcript-level pathway databases become available. Among the commonly down-regulated transcripts, we found enrichment in SRSF6-regulated and type 2 diabetes pathways, pointing to SRSF6-mediated splicing modifications in IFNα-treated EndoC-βH1 cells (Fig 4C, left panel). The same analysis run at gene level did not detect SRSF6-regulated pathways, but instead picked a few unrelated pathways (xenobiotics, linoleic acid metabolism, and muscle contraction, Supplemental Data 1). Hence, only RedRibbon captures meaningful splicing modifications (Fig 4B

and C). The down–up and up–down overlaps correspond to changes induced by IFNα that are not recapitulated by *SRFS6* knockdown and vice-versa, not discussed as we focus here on similarities. We next compared transcript patterns induced by IFNα in EndoC-βH1 cells and human islets (156,319 transcripts) and found strong similarity in down- and up-regulated transcripts (Fig 4B, right panel), with an overlap resembling Fig 1A, suggesting that the EndoC-βH1 cell line recapitulates alternative splicing of native human islets (isolated from organ donors) and is a valid model. Among the commonly down-regulated transcripts, pathway analysis identified enrichment of the *SRSF6* network, suggesting that SRSF6-regulated splicing modifications are present in human islets exposed to IFNα (Fig 4C, right panel). Again, gene-level analyses did not identify SRSF6-regulated splicing (Supplemental Data 1).

The transcripts commonly up-regulated in EndoC-βH1 cells by *SRSF6* silencing and IFNα exposure exhibit significant enrichment in interferon signalling, lysosomes, and apoptosis (Fig 5, left). Overlapping up-regulated transcripts between IFNα-exposed EndoC-βH1 cells and human islets showed additional enrichment in alternative splicing networks with a major role in beta cell signalling and apoptosis (Fig 5, right). Of note, 11 of the 16 enriched pathways in EndoC-βH1 cells (Fig 5, left) were present in human islets (Fig 5, right, highlighted in bold). Hallmarks of beta cells in type 1 diabetes were enriched, including the triad of MHC class I, stress pathways, and inflammation (toll-like receptors, interferon signalling). Altogether, these analyses illustrate the power of RedRibbon to compare very large transcript-level data and generate novel biological insight.

### Long-read RNA-sequencing

We next examined the performance of RedRibbon in comparing short- versus long-read RNA-seq. We exploited long-read RNA-seq of EndoC-βH1 beta cells exposed to the pro-inflammatory cytokines IFNγ and IL1β for 24 h (Thomaidou et al, 2021) and short-read RNA-

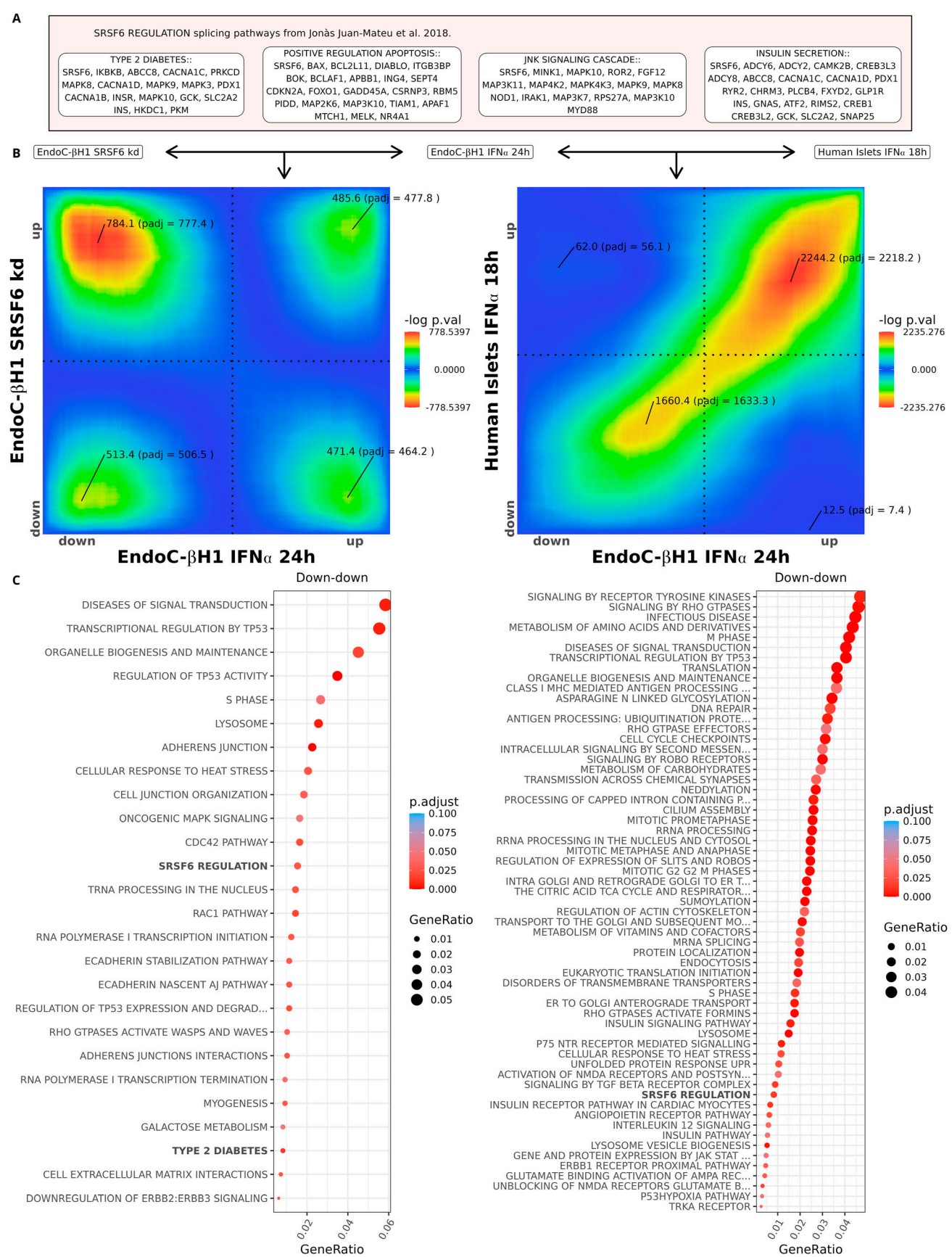

A

SRSF6 REGULATION splicing pathways from Jonàs Juan-Mateu et al. 2018.

**TYPE 2 DIABETES::**
SRSF6, IKBKB, ABCC8, CACNA1C, PRKCD
MAPK8, CACNA1D, MAPK9, MAPK3, PDX1
CACNA1B, INSR, MAPK10, GCK, SLC2A2
INS, HKDC1, PKM

**POSITIVE REGULATION APOPTOSIS::**
SRSF6, BAX, BCL2L11, DIABLO, ITGB3BP
BOK, BCLAF1, APBB1, ING4, SEPT4
CDKN2A, FOXO1, GADD45A, CSRNP3, RBM5
PIDD, MAP2K6, MAP3K10, TIAM1, APAF1
MTCH1, MELK, NR4A1

**JNK SIGNALING CASCADE::**
SRSF6, MINK1, MAPK10, ROR2, FGF12
MAP3K11, MAP4K2, MAPK4K3, MAPK9, MAPK8
NOD1, IRAK1, MAP3K7, RPS27A, MAP3K10
MYD88

**INSULIN SECRETION::**
SRSF6, ADCY6, ADCY2, CAMK2B, CREB3L3
ADCY8, ABCC8, CACNA1C, CACNA1D, PDX1
RYR2, CHRM3, PLCB4, FXYD2, GLP1R
INS, GNAS, ATF2, RIMS2, CREB1
CREB3L2, GCK, SLC2A2, SNAP25

seq of EndoC-βH1 cells exposed to the same cytokines for 48 h (Ramos-Rodríguez et al, 2019). The total number of transcripts being compared was 38,699. IFNγ and IL1β are implicated in late, adaptive immune processes in islets in type 1 diabetes (Eizirik et al, 2020). Strong overlap was observed in the up–up quadrant between the long and short transcripts (Fig 6), consistent with a nearly perfect overlap. In the highest up–up fold-change, short-reads tended to have a higher coordinate than long-reads, whereas the opposite was seen at the lower end of up–up-regulated transcripts. The absence of down–down overlap may be attributable to the shorter (24-h) exposure in the long-read study. Overlap of beta cell transcriptomes after cytokine exposure versus coming from individuals with type 1 diabetes is also limited to the up–up quadrant (Colli et al, 2020a).

# Discussion

The widespread use of omics technologies has exacerbated the need for new methods to analyse and compare diverse and ever larger datasets. Here we developed RedRibbon, a novel hypergeometric overlap method to remove the limitations of the original RRHO package (Plaisier et al, 2010), substantially increasing accuracy and size of the feasible analyses, and introducing novel data structures and algorithms. These improvements allow RedRibbon to leverage transcript-level quantification to detect overlapping signatures between two differential alternative splicing analyses. It features the capability to analyse lists one or two orders of magnitude longer without any loss of accuracy. The algorithms and data structures have been specifically tailored to be efficient (e.g., the bitset data structure allows to efficiently compute large set intersections using previously computed intersections). This new implementation goes beyond improving performance. First, gene or transcript overlap sets are provided for all four directions of regulation owing to the implementation of a two-sided test. The original R package did not report anticorrelated genes properly and the returned results were difficult to interpret (Cahill et al, 2018). The up–down and down–up quadrant enrichment lists are returned as the intersection of the set between the best coordinate and the quadrant corner.

Second, the accuracy of the localisation of the minimal $P$-value is improved over the grid method with an evolutionary algorithm. The grid method is pervasive among RRHO derivatives (Rosenblatt & Stein, 2014; Cahill et al, 2018; Thind et al, 2019) and other rank-based algorithms (Antosh et al, 2013). For an analysis of 20,000 genes, the original R package grid used a default step size of $\sqrt{20k}$ = 141, limiting the accuracy to this step size. The step parameter acts as a balance between speed and accuracy. A large step offers faster speed to the detriment of accuracy, and vice versa. Finding the right balance is difficult and there is no one-fits-all best value—the appreciation is left to the end user. As the error induced by the grid

method is proportional to the step size and the complexity of the original R package algorithm is $O(n(n/step)^2)$ for transcript lists, a small step is computationally expensive, whereas a large one gives inaccurate results leading to numerous false positives. Our evolutionary algorithm does not have this limitation and can accurately pinpoint the best $P$-value whatever the number of features analysed without impacting performance (Figs 2 and 3). The evolutionary algorithm complexity is $O(ipn)$ where $i$ is the number of iterations (default value 200) and $p$ is the population size (default value is $500 + \sqrt{n}$) giving a default parameters complexity of $O(n^{\frac{3}{2}})$. A downside is that it comes at the cost of non-determinism in the minimal $P$-value finding algorithm. We mitigated this by initializing the algorithm population with evenly spaced coordinates on the diagonal of the map. In the experiments presently performed, we did not detect any minimal $P$-value worse than the ones detected by the grid method and the returned overlap sets were always close to identity in case of non-determinism.

Third, the computation of the overlap map is decoupled from the minimal $P$-value search. Hence, locating minimal $P$-value coordinates is independent of visualization map resolution. This helps to optimize memory usage, something that is particularly important in the analysis of very long lists. Our minimal $P$-value search algorithm only keeps in memory for the grid algorithm the best coordinates, and for the evolutionary algorithm, the current population of coordinates, guaranteeing a very small memory footprint.

Eventually, a novel permutation scheme considering the correlation between genes is available to adjust the minimal $P$-value. This permutation scheme allows to correct the minimal $P$-value without having to re-run the whole differential analysis thousands of times while still considering the correlation between genes or transcripts. Doing so, it is possible to run a permutation scheme over long lists and many conditions as shown in Fig 2.

The package has been validated on synthetic datasets (Fig 3A and B) and previously reported RRHO results from Marselli et al (2020) (Fig 3C and D). The RedRibbon evolutionary algorithm detected synthetic dataset genes with a systematically better accuracy compared with the original algorithm. A similar difference in the number of detected genes was also present for real datasets of human islets in conditions related to type 2 diabetes, suggesting similar accuracy improvements. The differences are particularly marked when the overlap signal is diffuse (e.g., Fig 3C, down–down quadrant) as the step size misses the minimum, the surrounding $P$-values being very close in a large area. The observed differences are propagated in the pathway analyses. Hence, pinpointing the minimal $P$-value with accuracy is an important and unique feature of RedRibbon.

The package has been further applied to and validated for previously reported alternative splicing results in EndoC-βH1 cells and human islets (Juan-Mateu et al, 2018; Ramos-Rodríguez et al, 2019; Colli et al, 2020a; Thomaidou et al, 2021) (Figs 4–6). These analyses were done at the transcript level on lists comprising

**Figure 4. Transcript level analysis of SRSF6 regulated alternative splicing network.**
**(A)** Genes and pathways regulated by *SRSF6* as identified in Juan-Mateu et al (2018). **(B)** RedRibbon transcript level overlap maps comparing differential analyses of IFNα-treated and *SRSF6*-silenced EndoC-βH1 cells (left), and IFNα-treated EndoC-βH1 cells and IFNα-treated human islets (right). **(C)** Molecular Signature Database and canonical pathways enriched in overlapping down-regulated transcripts. The pathways known to be regulated by *SRSF6* are highlighted in bold. Left: IFNα-treated and *SRSF6*-silenced EndoC-βH1 cells. Right: IFNα-treated EndoC-βH1 cells and IFNα-treated human islets.

**Figure 5.  Molecular Signature Database and canonical pathways enriched in commonly up-regulated transcripts in beta cells after SRSF6 silencing and IFNα exposure.**
Left: pathways enriched in up-regulated transcripts in IFNα-treated and SRSF6-silenced EndoC-βH1 cells. Right: pathways enriched in up-regulated transcripts in IFNα-treated EndoC-βH1 cells and human islets. Pathways present in (left) are highlighted in bold in (right).

around 150,000 transcripts (see Supplemental Data 1), list lengths that are beyond the reach of the original R package. RedRibbon allowed running these analyses with accuracy and permutation-adjusted *P*-values in a matter of minutes. Our results suggest that SRSF6 splicing regulation transposes from EndoC-βH1 cells to human islets as the SRSF6 splicing pathways are enriched in both for down-regulated transcripts. Moreover, most of the up-regulated transcript pathways in EndoC-βH1 cells are recapitulated in human islets. The availability of human islets of Langerhans is limited, whereas EndoC-βH1 cells are readily available. The present analyses show that EndoC-βH1 cells recapitulate the alternative splicing patterns of human islets, making this cell line an appropriate model to study alternative splicing in human beta cells by deep sequencing, expanding beyond earlier validation studies of the EndoC-βH1 cells (Scharfmann et al, 2014; Hastoy et al, 2018; Tsonkova et al, 2018; Lawlor et al, 2019). The

comparison of long- and short-read RNA-seq datasets of cytokine-exposed EndoC-βH1 cells demonstrated remarkable overlap (Fig 6). RedRibbon validates that, despite differences in the size of the fragments being sequenced, long and short reads similarly capture splicing events.

The results obtained here show the importance of transcript level analyses to capture the effects of alternative splicing. We designed the SRSF6 regulatory pathway based on previous splicing analysis (Juan-Mateu et al, 2018), and it is only detectable at the transcript level. For other pathways, one of the challenges is that pathway databases are gene oriented. For those, transcript sets returned by RedRibbon can be converted to genes before pathway enrichment to compensate for the lack of transcript level pathway databases. Using this method, we obtained a large intersection between gene- and transcript-level pathway analyses and identified many new pathways for transcript level analyses. Obviously, the final enrichment is only

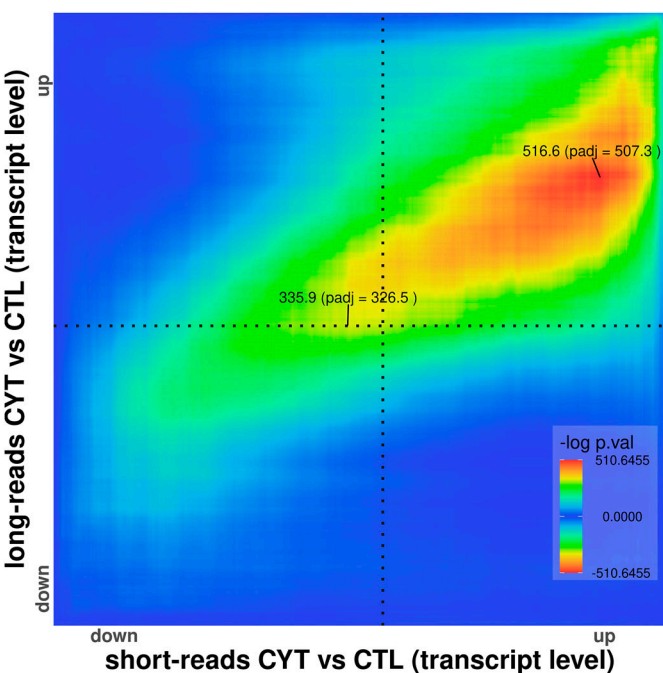

**Figure 6.  RedRibbon overlap between long- and short-read RNA-seq.**
The overlap map of long- and short-read transcripts of EndoC-βH1 beta cells exposed to IFNγ and IL1β shows a strong signal in the up–up but not in other quadrants.

as good as the pathway databases. Splicing network regulatory pathways may not be detected without specifically tailored databases, as done here for SRSF6. The creation of new transcript level pathway databases will enable refined alternative splicing analyses.

In conclusion, RedRibbon is a very useful novel tool to compare both gene-level and transcript-level differential analyses. Specifically, RedRibbon allows the detection of splicing networks. Using this tool, we documented large transcript-level similarities between EndoC-βH1 cells and human islets. Worth of note, the method is robust even when the cell types are not matched, as is the case here for immortalized beta cells and bulk human islets that contain around 50% beta cells. RedRibbon will be a very useful addition to the bioinformatic toolsets for the analysis and comparison of diverse ever bigger datasets.

# Materials and Methods

The RedRibbon package is a complete rewrite of the original package (Plaisier et al, 2010; Rosenblatt & Stein, 2014) with performance in mind and includes novel algorithms and data structures. RedRibbon enables the execution of a comprehensive RRHO analysis including the computation of adjusted P-values using gene correlation. It can be applied to lists or differential analyses that consist of millions of elements. A C library and an easy-to-use R package are provided. It implements new performant algorithms to find the minimal P-value, a novel adjusted minimal P-value computation algorithm, improved plots, and parallel execution.

## RedRibbon workflow

The input for RedRibbon is two lists of gene or transcript ranked statistics such as fold change or direction signed P-value (Fig 7A). From these, the minimal hypergeometric P-value coordinates are identified in the four quadrants of the level map using the original method (Plaisier et al, 2010)—called in this article "grid method" in reference to the grid-like traversal of the coordinate matrix—or our fast and accurate evolutionary algorithm (see below). Locating the minimal coordinates aims to split overlapping map quadrants into two areas, an enriched and a randomly ordered region. Optionally, the minimal P-value can be adjusted considering expression level correlation between genes or transcripts. The result is four transcript sets, one per quadrant. The enrichment result, the corrected P-values, and the four quadrants can be visualized in an overlap map.

Pathway enrichment analyses usually use overlapping genes in each quadrant. For transcript-level analyses, as most databases are gene centric, it is necessary to convert transcripts to genes. Gprofiler2 R package (Kolberg et al, 2020) and clusterprofiler R package (Yu et al, 2012; Wu et al, 2021) were used to do the enrichment analysis, respectively, for gene level and alternative splicing analyses. For alternative splicing analysis, we created five new pathways, namely *SRSF6* regulation, type 2 diabetes, positive regulation of apoptosis, insulin secretion, and JNK signalling. The list of spliced genes regulated by *SRSF6* were taken from Juan-Mateu et al (2018) (reproduced in Fig 4A) and have been converted into clusterProfiler-ready format.

## RedRibbon RRHO

### P-*value computation*
RedRibbon P-values can be computed with one- or two-sided or compatible with the original R module two-sided statistics. With $c$ being the number of genes in common for the coordinate $(i,j)$ in an RRHO map of size $n \times n$, the P-value is computed as follows:

$$pval = \begin{cases} 1 - CDF_{hyper}(c-1, i, j, n) & For\ one\ sided \\ 2 * \min\left(1 - CDF_{hyper}(c-1, i, j, n), CDF_{hyper}(c, i, j, n)\right) & For\ two\ sided \end{cases}$$

The two-sided test allows detecting enrichment both in correlated (up/up and down/down) and anticorrelated (up/down and down/up) genes. The anticorrelated genes were not reported by the original R Package. In addition, if the hypergeometric P-value comes from the lower tail of the distribution, they are negatively signed to distinguish depletion (anticorrelation) from enrichment (correlation).

### C *language implementation and R module*
RedRibbon has been split between a performant C library and an easy-to-use R module interfacing this library. RedRibbon has been optimised to be efficient regardless of the length of the lists given as input. The gene sets are represented by bit vectors allowing the use of CPU bit instructions. Bit vectors allow to efficiently compute set intersections, an essential operation for RRHO as it is done for each P-value computation. In addition, the intersected gene sets relative for two close coordinates on the RRHO map are very similar,

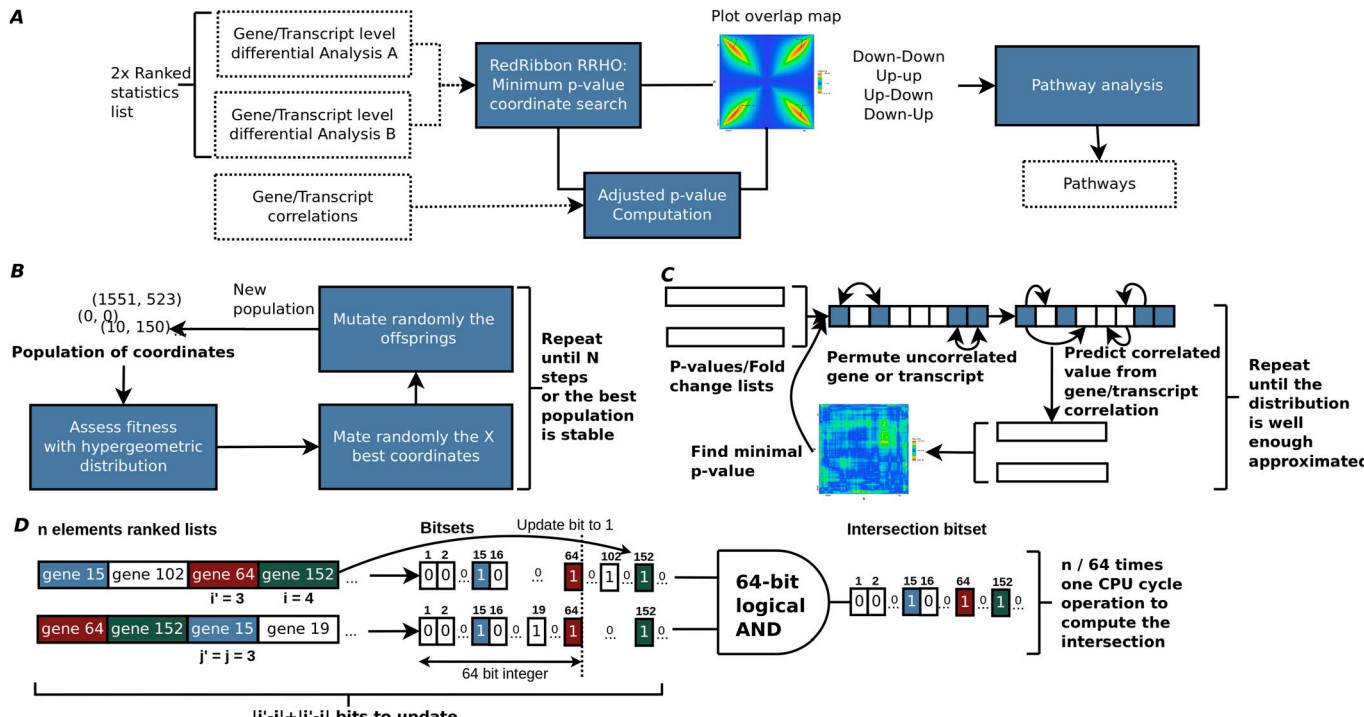

**Figure 7. RedRibbon rank–rank hypergeometric overlap (RRHO) workflow, algorithms, and data structures.**
**(A)** Transcript level differential analysis by RRHO. The RedRibbon RRHO package can handle very large data because of improved data structures and algorithms (see benchmark). Transcript level differential analyses can be overlapped with a permutation scheme to correct $P$-values. The overlap analysis is followed by a pathway analysis. **(B)** The evolutionary algorithm will find the minimal $P$-value among coordinates. The best fitness individuals of a population of coordinates are mated and then randomly mutated to obtain a new population. This process is repeated until stability is reached among the best population or a fixed number of steps. **(C)** Hybrid prediction–permutation method to compute the adjusted minimal $P$-value. A set of uncorrelated elements (genes, transcripts; shown in blue squares) is identified. Their value ($P$-value or fold change) is permuted. The remaining correlated elements of the lists are predicted from this set with a linear model. The minimal RRHO $P$-value is then computed for the two permutated lists. The operation is repeated a fixed number of times and the adjusted $P$-value assessed. **(D)** The bitset data structure to compute intersection of gene sets for the ranked lists at coordinate $(i,j)$. The ranked lists are converted into vectors of bits (bitsets) using the indexes of the genes. The bit at position k is set to one if gene k is before the index i or j in the ranked lists, otherwise the bit is set to zero. If a previous intersection has already been computed for coordinate $(i',j')$, the bitsets are only updated for the genes added or removed to reach the coordinate $(i,j)$. The intersection is computed with 64-bit logical AND operations.

most genes being in common. We leverage this similarity to decrease the operation numbers to compute the intersections by updating previously computed sets.

To improve the accuracy of small $P$-value computation which can be smaller than the smallest representable positive number by C language, *long double* type has been used for large lists rather than *double*. Doing so on an x86 platform, the smallest positive number becomes $3.36 \times 10^{-4932}$ instead of $2.23 \times 10^{-308}$. As $R$ does not support the *long double* type, we use the logarithm of the $P$-value to stay in the representable range for the $R$ package.

### Evolutionary algorithm to find the minimal $P$-value

Two minimal $P$-value search methods are implemented in RedRibbon: (1) the grid-based method used in the original RRHO implementation, and (2) an evolutionary algorithm-based method (Fig 7B). The latter interprets coordinates on the map as individuals of a population subject to selective pressure. The initial set of coordinates is chosen to be uniformly spaced on the diagonal of the RRHO map, a choice purposely reminiscent of the classical method to guarantee with a high probability that it remains at least as good as the classical

method. Next, a new generation is created by mating the coordinates and randomly introducing mutations to induce genetic diversity. This new population is then selected for the coordinates with the best fitness, measured with hypergeometric $P$-values and where lower is better. The process is repeated until a stable set of best coordinates (i.e., no newly added coordinates in best coordinate set) or a predefined number of generations is reached.

### Adjusted minimal $P$-values

The RRHO minimal $P$-value coordinate is selected among a large set of coordinates. For lists of N features, the minimal $P$-value coordinate is to be found in $N^2$ coordinates, each associated with one colour dot in the overlap map. Correcting the minimal coordinate $P$-values presents multiple challenges: (1) whereas the distribution for one specific coordinate is hypergeometric, the distribution of the minimal $P$-value coordinate is, to the best of our knowledge, unknown, (2) the features can be correlated, for example, from gene interaction, (3) the coordinate $P$-values are highly dependent as the hypergeometric $P$-value for two close coordinates is computed from sets with many elements in common; hence, the false

discovery rate correction assuming independence of variables is inadequate to compute a corrected minimal *P*-value, (4) permuting samples and redoing the full differential expression analyses many times as in Plaisier et al (2010) is very time consuming, unpractical and out of reach for very large datasets. Here, we introduce a new permutation scheme considering the correlation without redoing the differential analysis for each permutation (Fig 7C).

### Hybrid prediction–permutation method

RedRibbon-adjusted minimal *P*-values are computed using an efficient hybrid prediction–permutation method (HPP) to assess the null distribution of the minimal coordinate *P*-value. When the features are independent, the HPP method is strictly equivalent to a permutation of the feature lists. The HPP method divides the features in two disjoint sets, namely the predictor and predicted sets. The predictor set is composed of features which are not able to mutually predict each other. Each predicted set feature value is predictable from the predictor set feature values or another predicted feature. A HPP permutation is generated by first permuting the original predictor set values and then predicting the other set features from these.

For a list of fold changes $FC_i$, the predicted values are computed with a linear model:

$$\log FC_y = \beta_1 \log FC_x$$

where the gene or transcript $x$ is in the predictor set and $y$ belongs to the predicted set and the $\beta_1$ coefficient is estimated from another linear model over expression levels $\log Expr_y = \beta_1 \log Expr_x + \beta_0$. The latter is justified as for two samples we have $\log FC_y = \log Expr_y^{(1)} - \log Expr_y^{(2)} = \beta_1 \log Expr_x^{(1)} + \beta_0 - \beta_1 \log Expr_x^{(2)} - \beta_0 = \beta_1 \log FC_x$. The linear model parameters are estimated with ordinary least square method from the expression matrices.

For a list of *P*-values $P_i$, the predicted values are computed from the expression correlation coefficient $r$ relative to a predictor and a random value $\widehat{P}$ generated from the distribution of all values in the original list:

$$P_y = |r|P_x + (1 - |r|)\widehat{P}$$

where $P_x$ is in the predictor set and $P_y$ belongs to the predicted set.

This formula assumes a linear effect between the value and the correlation coefficient. This effect is modelled by the term $|r|P_x$. In case of standardised predictor and predicted variables, r is the best linear ordinary least square estimator. The term $(1-|r|)\widehat{P}$ is a bootstrap estimate before taking into account the correlation of variables. The whole formula guarantees that the estimated value is equal to the predictor value if $|r|$ is one, whereas an r close to zero gives a bootstrapped random value. Consequently, the model can be seen as a finite mixture between the predictor and a bootstrap variable weighted with the correlation coefficient.

### Beta distribution fitting

The HPP method is repeated several times to obtain a list of minimal *P*-values. To limit the number of HPP iterations (around 100), a beta distribution is fitted on the HPP *P*-values, and the goodness-of-fit is assessed with a Kolmogorov–Smirnov test. If the goodness-of-fit test is verified, the threshold for 0.05 significance is given by moment methods fitted beta $CDF^{-1}(0.05)$ and the adjusted

*P*-value is $0.05/(C\widehat{DF}(-1)(0.05))*pvalue$. If the goodness-of-fit test rejects the hypothesis of beta distribution, the threshold is computed from the empirical cumulative distribution function.

### Computing the parameters of linear regressions and selecting HPP predictors

In case of fold change lists, for each gene, the best gene predictor is selected among the linear regression models with significant $\beta_1$. The minimal mean squared error model is used as fold change predictor. A gene is put in the HPP predictor set if it is the best predictor for another gene and is not itself predicted from another gene. For one gene, all linear models are computed at once with an efficient ordinary least square based on Householder reflections QR decomposition. For *P*-value lists, the best significant $|r|$ is selected.

### Data structures and algorithm

The computation of hypergeometric enrichment requires the computation of many set intersections. A bitset data structure is used to represent the sets (Fig 7D). The bitset is a C array of 64 bits unsigned integer allowing to intersect 64 elements in one computer cycle with a binary "and" instruction. This data structure divides the number of operations for the intersection by 64 and lowers the CPU cache pressure (Drepper, 2007) reducing the RAM storage by 32, the integer bit size of R language.

In addition, the intersection algorithm makes use of the previous intersection computation to reduce the number of updates to sets to intersect as the sets for the coordinate $(i + b, j)$ are the same as for previously computed $(i, j)$ set except for $b$ elements.

### Synthetic gene sets and accuracy measurements

To assess the accuracy of our method, synthetic sets of genes have been generated. The aim of these datasets is to assess the TPR (i.e., sensitivity), TNR (i.e., specificity), and the accuracy in a setting where the overlapping genes are known. The set of gene sets is composed of 192 artificial gene sets of 19,962 genes with 200 genes overlapping in the top 300 down-regulated genes.

### Transcriptomes and differential analyses

RNA-seq data that we previously generated were used to assess RedRibbon performance and its ability to generate new and accurate results. The transcriptomes come from human islets from type 2 diabetic and nondiabetic donors, the latter exposed or not to palmitate and high glucose (Marselli et al, 2020) or IFNα (Gonzalez-Duque et al, 2018; Colli et al, 2020a), and the human beta cell line EndoC-βH1 exposed to IFNα, or silenced for splicing factor *SRSF6* (Juan-Mateu et al, 2018).

Quality control and trimming were done with fastp 0.19.6. The bulk RNA-seq fastq were quantified with Salmon 1.4.0 (Patro et al, 2017) using the parameters *−seqBias −gcBias −validateMappings* with GENCODE v36 (Frankish et al, 2019) as the genome reference. Differential expression analyses were done with DESeq2 1.28.1 (Love et al, 2014).

**RedRibbon R package compatibility with the original implementation**

To facilitate reanalysis of existing datasets, the RedRibbon R package provides a compatibility mode. The original *RRHO* R function has been rewritten using the new algorithms of RedRibbon. Hence, the existing pipelines can be improved for accuracy and performance just by substituting the library inclusion with close to no code editing.

# Data Availability

All raw sequencing data are available at the NCBI Gene Expression Omnibus (GEO; https://www.ncbi.nlm.nih.gov/geo/) under accession numbers GSE159984, GSE133218, GSE137136, GSE98485, GSE148058, GSE108413, and GSE167223. The C libraries and R package code are open to the community with a permissive licence (GPL3) and available for download from GitHub https://github.com/antpiron/RedRibbon, https://github.com/antpiron/ale and https://github.com/antpiron/cRedRibbon. The supplementary material (Supplemental Data 2) provides examples how to use RedRibbon on synthetic and real datasets. All the code and processed sequencing data generated in this study to generate results and figures are available on zenodo Piron et al (2023).

# Supplementary Information

# Acknowledgements

We thank Xiaoyan Yi, ULB Center for Diabetes Research, for testing the *R* package. This work has been supported by the European Union's Horizon 2020 research and innovation program T2DSystems under Grant Agreement No. 667191, the Fonds National de la Recherche Scientifique (FNRS), the Walloon Region SPW-EER Win2Wal project BetaSource, Belgium, the Francophone Foundation for Diabetes Research (FFRD, that is sponsored by the French Diabetes Federation, Abbott, Eli Lilly, Merck Sharp & Dohme and Novo Nordisk), the FWO and FRS-FNRS under the Excellence of Science (EOS) programme (Pandarome project 40007487), and the Innovative Medicines Initiative 2 Joint Undertaking under grant agreements 115797 (INNODIA) and 945268 (INNODIA HARVEST). This latter joint undertaking receives support from the Union's Horizon 2020 Research and Innovation Programme and the European Federation of Pharmaceutical Industries and Associations, JDRF, and The Leona M. and Harry B. Helmsley Charitable Trust. DL Eizirik is also supported by grants from Welbio–FNRS, Belgium (WELBIO-CR-2019C-04), NIH-HIRN (5U01DK127786-02), USA, and the Innovate2CureType1-Dutch Diabetes Research Foundation (DDRF) Grant 2018.10.002. F Szymczak is supported by a Research Fellow (Aspirant) fellowship from the FNRS, Belgium (FC 038603). A Piron is supported by the David and Alice Van Buuren Fund, the Jaumotte-Demoulin Foundation, the Héger-Masson Foundation, the Wiener-Anspach Foundation, and the FNRS.

### Author Contributions

A Piron: conceptualization, data curation, software, validation, investigation, visualization, methodology, and writing—original draft, review, and editing.

F Szymczak: conceptualization, data curation, software, validation, and methodology.
T Papadopoulou: conceptualization, data curation, software, validation, investigation, visualization, and writing—review and editing.
MI Alvelos: conceptualization, formal analysis, and validation.
M Defrance: conceptualization and supervision.
T Lenaerts: conceptualization, supervision, and methodology.
DL Eizirik: conceptualization and supervision.
M Cnop: conceptualization, resources, formal analysis, supervision, funding acquisition, validation, project administration, and writing—original draft, review, and editing.

### Conflict of Interest Statement

The authors declare that they have no conflict of interest.

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
