## [Reviewer comments · Life Science Alliance]

Life Science Alliance

RedRibbon: A new rank-rank hypergeometric overlap for gene and transcript expression signatures

Anthony Piron, Florian Szymczak, Theodora Papadopoulou, Maria Alvelos, Matthieu Defrance, Tom Lenaerts, Decio Eizirik, and Miriam Cnop

DOI: <https://doi.org/10.26508/lsa.202302203>

Corresponding author(s): Anthony Piron, Université Libre de Bruxelles

Review Timeline:	Submission Date:	2023-06-07
	Editorial Decision:	2023-08-08
	Revision Received:	2023-11-01
	Editorial Decision:	2023-11-17
	Revision Received:	2023-11-28
	Accepted:	2023-11-29

Transaction Report:

August 8, 2023

Re: Life Science Alliance manuscript #LSA-2023-02203-T

ANTHONY PIRON

Dear Dr. Piron,

Thank you for submitting your manuscript entitled "RedRibbon: A new rank-rank hypergeometric overlap pipeline to compare gene and transcript expression signatures" to Life Science Alliance. The manuscript was assessed by an expert reviewer, whose comments are appended to this letter. We invite you to submit a revised manuscript addressing the Reviewer comments.

When submitting the revision, please include a letter addressing the reviewer comments point by point.

Thank you for this interesting contribution to Life Science Alliance. We are looking forward to receiving your revised manuscript.

Sincerely,

B. MANUSCRIPT ORGANIZATION AND FORMATTING:

Reviewer #1 (Comments to the Authors (Required)):

The authors present the RedRibbon software that was developed to overcome some of the shortcomings that remain for the original RRHO (Rank-Rank Hypergeometric Overlap) approach developed for microarray data (by Plaisier et al. 2010, NAR) which compares 2 lists of differentially expressed genes.

The advance the authors present avoids some of the numerical and performance issues associated with the original RRHO approach (underflow issues, slow run time on large datasets,...) with performance was evaluated on both synthetic and experimental datasets.

The results presented support the claims made in the manuscript.

Areas to improve:

Please check statement at the top of page 4 beginning "Despite the progress..." and clarify what is meant, as there are plenty of methods for differential transcript analysis (any gene-level method will work on transcript counts).

Regarding directional testing, the method of Wu et al. (2010) (Bioinformatics, 26(17):2176-82, <https://pubmed.ncbi.nlm.nih.gov/20610611/>) allows directional testing between 2 sets and can be applied to either microarray data (as per the original publication) or RNA-seq (as implemented in the limma software (roast), along with a faster implementation (fry)). It would be good to position the work presented in relation to this and other similar approaches.

It was unclear how the transcript-level results are linked to external signature databases in Figure 4c and Figure 5 (as the authors point out in the Introduction, current pathway databases are gene-centric, so it would be good to explain how transcripts are mapped to genes in these analyses somewhere in the manuscript).

The RedRibbon user guide was quite brief, and I wondered if it would benefit from further examples of application of the software? Has it been applied to further datasets? Are there any long-read RNA-seq datasets which could be analysed by this method (they may allow more accurate quantification of transcript-level expression changes than short-read RNA-seq)?

There is very little detail on the data structures in the methods, a topic which is repeatedly mentioned in the article as something that has been improved in this work. Perhaps this can be either toned down throughout if it is only a minor thing, or the methods can be expanded with a figure added to explain this important enhancement in more detail.

Check expression p19 'RedRibbon allows to do a full...' (missing word/s?)

Check for consistency throughout: P-value (italicised versus non-italicised P)

Missing spaces - page 23: 4.4.1 Hybrid prediction-permutation methodRedRibbon

- page 24: 4.4.3 Computing the parameters of linear regressions and selecting HPP predictorsIn

- Please provide the full link to the code (the zenodo number given isn't given) <https://zenodo.org/record/7636157>

Reviewer #1 comments:

The authors present the RedRibbon software that was developed to overcome some of the shortcomings that remain for the original RRHO (Rank-Rank Hypergeometric Overlap) approach developed for microarray data (by Plaisier et al. 2010, NAR) which compares 2 lists of differentially expressed genes.

The advance the authors present avoids some of the numerical and performance issues associated with the original RRHO approach (underflow issues, slow run time on large datasets,...) with performance was evaluated on both synthetic and experimental datasets.

The results presented support the claims made in the manuscript.

We thank the Reviewer for the careful review of our paper and the positive and constructive feedback.

Areas to improve:

Please check the statement at the top of page 4 beginning "Despite the progress..." and clarify what is meant, as there are plenty of methods for differential transcript analysis (any gene-level method will work on transcript counts).

Thank you for this valid comment. We have now clarified in the introduction that several differential transcript analysis tools exist. No tools were available, however, to compare two distinct differential transcript analyses.

Regarding directional testing, the method of Wu et al. (2010) (Bioinformatics, 26(17):2176-82, <https://pubmed.ncbi.nlm.nih.gov/20610611/>) allows directional testing between 2 sets and can be applied to either microarray data (as per the original publication) or RNA-seq (as implemented in the limma software (roast), along with a faster implementation (fry)). It would be good to position the work presented in relation to this and other similar approaches.

Thank you for your suggestion to position our work in relation to ROAST. ROAST is an interesting, focused gene set enrichment method with an original P -value correction method using rotations to account for gene correlation. This method is, to our understanding, not comparable to RedRibbon nor transposable to two differential analyses. Instead, it compares one differential analysis against a focused gene set.

The differential studies under examination may not be expressible as a linear model. RedRibbon has a generic setup requiring a statistic that can be ranked, such as fold-changes or P -values (the model to generate them can even be unknown).

The ROAST statistics expect for the comparison a gene set and not a full differential analysis including non-significant genes. To this end, one differential analysis can be converted to a gene list using a significance threshold (e.g., P -adjusted ≤ 0.05). It hence does not have the threshold-freeness of RRHO.

We have now included in the Introduction (page 3) the focused gene set testing by ROAST, to position RedRibbon as a threshold-free method to compare two differential analyses.

It was unclear how the transcript-level results are linked to external signature databases in Figure 4c and Figure 5 (as the authors point out in the Introduction, current pathway databases are gene-centric, so it would be good to explain how transcripts are mapped to genes in these analyses somewhere in the manuscript).

We thank the Reviewer for this valid comment. RedRibbon generates transcript lists for each regulation quadrant when applied to two differential transcript expression analyses. To perform pathway analyses, we replaced the transcript identifiers with their corresponding gene identifiers using GENCODE annotation (GRch38 version 37). Multiple transcripts in the same quadrant or in different quadrants may be mapped to the same gene. The loss of this multiplicity information is inherent to using gene-centric pathway databases and will be solved when transcript-level pathway databases become available. We have included the information on how transcripts were mapped to genes (and the loss of information that comes with it) in the Results section describing Figure 4c and Figure 5.

The RedRibbon user guide was quite brief, and I wondered if it would benefit from further examples of application of the software? Has it been applied to further datasets? Are there any long-read RNA-seq datasets which could be analysed by this method (they may allow more accurate quantification of transcript-level expression changes than short-read RNA-seq)?

Following the Reviewer's excellent suggestion, we did a major overhaul of the user guide. We enhanced and clarified the synthetic data set introduction with more context and a description of each parameter used. We also included in the user guide more diverse omics datasets. First, we obtained long-read RNA-seq data of cytokine-exposed EndoC- β H1 cells and compared these against our short-read RNA-seq. This interesting comparison identified a strong signal in the up-upregulation quadrant, consistent with a nearly perfect overlap. This new analysis has also been included in the main text (page 10) and as a new Figure 6. Second, we used RedRibbon to compare proteomic and transcriptomic datasets, further illustrating the flexibility and robustness of the tool in terms of dataset comparisons. This comparison has been included in the supplemental R vignette.

There is very little detail on the data structures in the methods, a topic which is repeatedly mentioned in the article as something that has been improved in this work. Perhaps this can be either toned down throughout if it is only a minor thing, or the methods can be expanded with a figure added to explain this important enhancement in more detail.

We thank the Reviewer for this valid comment. A new panel D has been added to Figure 7 to illustrate the bitset data structure and the optimization of the intersection algorithm described in section 4.5. The Figure legend and the Methods section have been adapted accordingly. This new panel D in Figure 7 illustrates the bit level parallelism and the use of previously computed intersection. These two points are central to achieve the required performance for transcript analyses.

Check expression p19 'RedRibbon allows to do a full...' (missing word/s?)

We corrected the wording of the sentence as "RedRibbon enables the execution of a comprehensive RRHO analysis including the computation of adjusted *P*-values using gene correlation. It can be applied to lists or differential analyses that consist of millions of elements."

Check for consistency throughout: P-value (italicised versus non-italicised P)

We uniformized the *P*-value notation. The italicized *P* version "*P*-value" is now used for the whole manuscript. This notation is commonly used in Life Science Alliance publications.

Missing spaces - page 23: 4.4.1 Hybrid prediction-permutation methodRedRibbon

- page 24: 4.4.3 Computing the parameters of linear regressions and selecting HPP predictorsIn

We apologize for this; in the conversion of MS Word to PDF some spaces disappeared. This has been corrected.

- *Please provide the full link to the code (the zenodo number given isn't given)*
<https://zenodo.org/record/7636157>

The full link is now provided in "5. data access" section (<https://zenodo.org/records/7585784>).

November 17, 2023

RE: Life Science Alliance Manuscript #LSA-2023-02203-TR

Dr. Anthony Piron
Université Libre de Bruxelles
ULB Center for Diabetes Research
Route de Lennik 808 (U.L.B. CP618)
Brussels (Anderlecht) 1070
Belgium

Dear Dr. Piron,

Thank you for submitting your revised manuscript entitled "RedRibbon: A new rank-rank hypergeometric overlap for gene and transcript expression signatures". We would be happy to publish your paper in Life Science Alliance pending final revisions necessary to meet our formatting guidelines.

- please address the Reviewer's remaining minor comment
- please upload a clean manuscript without tracking changes
- please note that titles in the system and on the manuscript file must match
- please make sure the manuscript sections are aligned with LSA's formatting guidelines: please separate the Figure Legends and Supplemental Figure Legends into separate sections
- Regarding the Supplementary file, "Vignette S1": this additional information can remain as Supplementary Material (and should be referred to as Supplementary Material), but it should not have a unique set of authors, and should not have its own Abstract, as this is not a separate publication. Feel free to reach out via email if I am misunderstanding the intention of this section.

A. FINAL FILES:

B. MANUSCRIPT ORGANIZATION AND FORMATTING:

Sincerely,

Reviewer #1 (Comments to the Authors (Required)):

The authors have carefully addressed the questions raised during review.

Minor:

Check the expression in the sentence beginning on line 90. It might read better as "Multiple differential analysis tools can be applied to transcript-level data (..."

November 29, 2023

RE: Life Science Alliance Manuscript #LSA-2023-02203-TRR

Dr. Anthony Piron
Université Libre de Bruxelles
ULB Center for Diabetes Research
Route de Lennik 808 (U.L.B. CP618)
Brussels (Anderlecht) 1070
Belgium

Dear Dr. Piron,

Thank you for submitting your Methods entitled "RedRibbon: A new rank-rank hypergeometric overlap for gene and transcript expression signatures". It is a pleasure to let you know that your manuscript is now accepted for publication in Life Science Alliance. Congratulations on this interesting work.

DISTRIBUTION OF MATERIALS:

Again, congratulations on a very nice paper. I hope you found the review process to be constructive and are pleased with how the manuscript was handled editorially. We look forward to future exciting submissions from your lab.

Sincerely,
